# Modeling the Reduction of *Salmonella* spp. on Chicken Breasts and Wingettes during Scalding for QMRA of the Poultry Supply Chain in China

**DOI:** 10.3390/microorganisms7060165

**Published:** 2019-06-06

**Authors:** Xingning Xiao, Wen Wang, Xibin Zhang, Jianmin Zhang, Ming Liao, Hua Yang, Qiaoyan Zhang, Chase Rainwater, Yanbin Li

**Affiliations:** 1College of Biosystems Engineering and Food Science, Zhejiang University, Hangzhou 310058, China; xingningxiao@126.com; 2MOA Laboratory of Quality & Safety Risk Assessment for Agro-products (Hangzhou), State Key Laboratory for Quality and Safety of Agro-products (in prepared), Institute of Quality and Standard of Agricultural Products, Zhejiang Academy of Agricultural Sciences, Hangzhou 310021, China; yanghua@zaas.ac.cn (H.Y.); yanyan0014@163.com (Q.Z.); 3College of Food Science and Engineering, Shandong Agricultural University, Taian, Shandong 271018, China; xbzhang84@163.com; 4New Hope Liuhe Co., Ltd., Beijing 266000, China; 5College of Veterinary Medicine, South China Agricultural University, Guangzhou 510642, China; junfeng-v@163.com (J.Z.); mliao@scau.edu.cn (M.L.); 6Department of Industrial Engineering, University of Arkansas, Fayetteville, AR 72701, USA; cer@uark.edu; 7Department of Biological & Agricultural Engineering, University of Arkansas, Fayetteville, AR 72701, USA

**Keywords:** poultry scalding, *Salmonella* reduction, probability distribution, Weibull model

## Abstract

The objective of this study was to develop predictive models for describing the inoculated *Salmonella* reductions on chicken during the scalding process in China. *Salmonella* reductions on chicken breasts at a 100 s treatment were 1.12 ± 0.07, 1.38 ± 0.01, and 2.17 ± 0.11 log CFU/g at scalding temperatures of 50, 60 and 70 °C, respectively. For chicken wingettes, 0.87 ± 0.02, 0.99 ± 0.14 and 1.11 ± 0.17 log CFU/g reductions were obtained at 50, 60 and 70 °C after the 100 s treatment, respectively. Greater bacterial reductions were observed on chicken breasts than on chicken wingettes (*p* < 0.05). A logistic (−1.12, 0.06) distribution could describe the bacterial reductions on chicken breasts at 50–60 °C. Weibull, exponential and log-linear models were compared for describing the bacterial reduction on chicken breasts at 70 °C and the Weibull model showed the best fit as indicated by the pseudo-*R*^2^, root mean square error (RMSE) and standard error of prediction (SEP) values. For chicken wingettes, a logistic (−0.95, 0.07) distribution could be used to describe the bacterial reduction at 50–70 °C. The developed predictive models could provide parts of the input data for microbial risk assessment of the poultry supply chain in China.

## 1. Introduction

*Salmonella* spp. are Gram-negative facultative intracellular zoonotic pathogens [1,2]. It was estimated there are 80.3 million cases of foodborne illnesses associated with *Salmonella* globally each year, posing a large public health and economic burden [3,4]. Poultry products are a frequent source of *Salmonella* contamination [1,3,5]. In 2016, chicken production was 1.72 million tons in China, while it was reported that the poultry products at retail markets were associated with a high prevalence of *Salmonella* [2,6,7,8,9]. Even with the most rigorous hygienic operations and full cooking, contaminated products may still lead to kitchen cross-contamination, which is a potential threat to human health.

According to the operating procedure of chicken slaughtering (GB/T 19478-2004), flow chart of the poultry slaughtering chain is shown in Figure 1. Scalding is the step using hot water immersion to facilitate the removal of feathers and reduce the pathogenic bacteria on meat during slaughtering [10,11]. There are two types of scalding process. One is the ‘hard scald’ with water temperatures from 60 to 66 °C and immersion times of 45 to 90 s, and another is the ‘soft scald’ with lower water temperatures from 51 to 54 °C and longer immersion times of 120 to 210 s [12,13]. Most US commercial poultry slaughter houses choose the hard scalding method in order to remove the outer skin surface cuticle layer on chicken carcasses [13]. Besides, it is popular to use chemical additives in scalding water to enhance bacterial reduction [14]. Scalding parameters in slaughter houses in China are different from other countries. In China, due to the difference of facility scale and age of broilers, the scalding water temperatures vary from 50 to 70 °C and scalding times are from 60 to 100 s, and no chemical additives are allowed to be used in scalding water. Yellow-feathered broiler and white-feathered broiler are two of the most popular chicken species in China. Compared to white-feathered broiler, yellow-feathered broiler with a yellow skin is easily affected by temperature during scalding. Generally, the scalding water temperatures of yellow-feathered broiler vary from 50 to 60 °C and the scalding water temperatures of white-feathered broiler vary from 60 to 70 °C according to the onsite investigation.

Predictive microbiology has been used as an important tool to improve food safety by developing mathematical models to quantitatively predict the growth or death of microorganisms under prescribed environmental conditions during food processing. In particular, predictive microbiological models can be used in a quantitative microbial risk assessment (QMRA) model to estimate the level of microorganisms in food at the time of consumption [15]. Therefore, modeling of *Salmonella* reduction during scalding is critical to QMRA of *Salmonella* for the whole poultry supply chain in China. In previous studies, probability distributions, such as Log Uniform, Normal, and BetaGeneral distributions, fitted by the sampling data randomly collected before and after scalding in slaughter houses, have been used to describe the *Campylobacter* reductions on chicken carcasses, while these data are not available for the QMRA of *Salmonella* in China [16,17,18]. In previously published laboratory scale tests, the scalding models were developed based on a long time (greater than 2 min) scalding processes while scalding times less than 2 min were commonly used in China [10,19,20]. Predictive models of *Salmonella* with thermal inactivation on chicken breasts based on a short time (2 min) were investigated, while the QMRA models need to be validated due to the differences of physical-chemical properties between chicken breasts and skins [21]. Experiments with skinless chicken breasts are more related to time or temperature decontamination studies, while skin injuries will happen during scalding, leading to skinless chicken meat exposure to the hot water. Very few predictive models or probability distributions are available for the description of *Salmonella* reduction during scalding for QMRA of the poultry supply chain in China.

Therefore, the objectives of this study were to investigate the reductions on skinless chicken breasts and skin-on chicken wingettes within the water temperatures of 50–70 °C and the treatment times of 0–100 s in lab-scale test, and to develop probability distribution or predictive model for describing the bacterial reduction. This study could provide some input data for a QMRA model of *Salmonella* in the poultry supply chain in China.

## 2. Materials and Methods

### 2.1. Bacterial Inoculum

Five *Salmonella* strains belonging to different serotypes (Stanley BYC12, Indiana HZC10, Typhimurium YXC1, Thompson LWC10, Kentucky CBC2) isolated from chicken carcasses were obtained from Dr. Liao’s laboratory at South China Agricultural University, Guangzhou, China. The bacterial strains were maintained in brain heart infusion broth (BHI, Becton Dickinson (BD), Franklin Lakes, NJ, USA) containing 20% (v/v) glycerol at −80 °C. Each strain was separately incubated in BHI at 37 °C for 24 h and cultured to approximately 9 log CFU/mL. Equal volumes of each culture suspension were mixed to obtain a five-strain mixture of *Salmonella*. Appropriate 10-fold dilutions in sterile phosphate buffered saline (PBS, Sigma, St. Louis, MO, USA) were made and plated on Xylose Lysine Tergitol-4 (XLT4, BD) agar to determine the cell number in the inoculums.

### 2.2. Preparation and Inoculation of Chicken Samples

Skinless chicken breasts (weight: 25 ± 3 g, size: 5 × 3 × 2 cm) and skin-on chicken wingettes (weight: 40 ± 3 g, size: 7 × 4 × 1.7 cm) were purchased from a local supermarket and stored in a freezer at −20 °C. After thawing overnight at 4 °C, the samples were exposed to ultraviolet (UV) light for 30 min in a biosafety cabinet (Thermo Fisher 1389, Waltham, MA, USA) to decontaminate initial microbial contamination. Control experiments showed that no *Salmonella* was initially present on the three chicken breasts and three wingettes, respectively. Then the samples were submerged into *Salmonella* suspensions for 30 min and transferred to plastic plates for another 30 min to allow bacterial attachment, both of which were performed at ambient temperature.

### 2.3. Scalding Treatments

According to the operation of the poultry slaughter houses in China, the ratio of chicken to scalding water is 1:4 (w/v), thus 30 chicken breast samples (a total of 750 g) were submerged into 3 L water and thirty chicken wingette samples (a total of 1200 g) were submerged into 4.8 L water to mimic the processing practice. Scalding treatments at 50, 60 and 70 °C were carried out in a laboratory water bath (TX150, Grant, Royston, UK) equipped with a digital thermometer (34970A, Agilent, Santa Clara, CA, USA) to monitor both temperatures of water and chicken meat. Thirty inoculated chicken breasts were submerged into the water bath and three samples were removed for processing every 10 s within a 100 s treatment period. The samples were individually added to sterile stomacher bags (Seward, London, UK) containing 25 mL buffered peptone water (BPW, BD, Franklin Lakes, NJ, USA) and homogenized for 1 min in a Model 400 food stomacher (Seward, London, UK). CFU levels were determined after appropriate dilutions in BPW were plated as described below. The initial inoculation load of chicken breasts and wingettes was 6.6 ± 0.1 log CFU/g and 6.3 ± 0.1 log CFU/g, respectively. The bacterial population on chicken samples at *t* = 0 was determined using the inoculated samples without scalding treatment. The treatments at 25 °C were conducted to determine the effect of washing at room temperature on bacterial reduction. Each treatment was repeated three times on different days using duplicated plates for each sample. The same protocol of scalding treatments was used for chicken wingettes.

To determine the temperature profiles of scalding water and the surface of chicken breasts and wingettes during treatments, a digital thermometer with K-type thermocouples was used The thermocouples (34970A, Agilent, Santa Clara, CA, USA) were inserted on the top surface of samples to measure the surface temperatureTemperatures were recorded every 10 s for the 100 s treatments.

### 2.4. Bacterial Enumeration

A preliminary test concerned the recovery of thermally injured *Salmonella*. Bacterial reductions at 50 s and 100 s treatments on the three chicken breasts and wingettes were tested. XLT4 agar was compared with Tryptic Soy agar (TSA, BD, Franklin Lakes, NJ, USA) in plate counting to check whether there were heat injured bacteria that will grow after adjusting. Only less than 0.1 log CFU/g difference was observed, which was not significant (*p* > 0.05) as determined by analysis of variance (ANOVA). Therefore, populations of *Salmonella* were selectively enumerated on XLT4 agar by the spiral plating method. The homogenates were serially 10-fold diluted in BPW and a 50 µL portion of appropriate dilutions was plated in duplicate onto the XLT4 agar using a spiral plater (WASP 2, Don Whitley Scientific, Shipley, UK). The plates were incubated at 37 °C for 16 h. Colonies on XLT4 agar plates were enumerated by a ProtoCOL 3 automated colony counter (Synbiosis, Cambridge, UK). The limit of detection was one colony for 50 μL sample. The viable bacterial populations on samples were expressed as CFU/g.

### 2.5. Distribution Fitting

Presently, two mathematical approaches, the deterministic calculation and the probabilistic evaluation, have been used in risk assessment. The deterministic method is popular because it is simple to conduct and understand, while the estimates can be unrealistic and the uncertainty cannot be quantified. The probabilistic method has been preferred in the risk calculation [22]. Bacterial reductions showed a large variation because of multiple uncertainties that were involved, as well as the inherent errors in microbial collection from surfaces and enumeration techniques. Hence, in cases for which no significant differences of the bacterial reductions with scalding times or temperatures were observed, probability distributions were defined based on the Kolmogorov–Smirnov test to describe the uncertainty and variability of bacterial reductions using @RISK 7.5 software (Palisade, Newfield, NY, USA).

### 2.6. Model Development and Evaluation

When a significant difference of the bacterial reductions with scalding times was found, the bacterial reduction curve was fitted with the Weibull, exponential and log-linear models, respectively.

The Weibull model is based on the assumption that bacterial resistance to heat treatment varies from one cell to another [23]. The model has gained popularity for its simplicity and flexibility and is displayed in Equation (1):(1)$$\log N_{t}=\log N_{0}-{\left(\frac{t}{\beta }\right)}^{\alpha }$$
where *N_t_* is the bacterial population at time *t* (CFU/g), *N*_0_ is the initial bacterial population (CFU/g), and *β* and *α* are temperature-dependent coefficients that represent the scale and shape factors of the model, respectively. For *α* = 1, a linear curve is obtained, *α* > 1 describes a convex curve (shoulder) and indicates that the remaining cells continue to die, and *α* < 1 describes a concave curve (tailing) and indicates that the remaining cells have the ability to adapt to the applied stress.

The exponential model has been used in describing and predicting inactivation kinetics and different destruction rates [20]. The model is displayed in Equation (2):(2)$$\log N_{t}=\log N_{0}+b\times e^{\left(-t/c\right)}$$
where *N_t_* is the bacterial population at time *t* (CFU/g), *N*_0_ is the initial bacterial population (CFU/g), and *b* and *c* are coefficients derived from the regression analysis.

A log-linear model assumes a homogeneous bacterial resistance to heat treatment and has been frequently used for bacterial survival curve fitting in thermal process [24,25].
(3)$$\log N_{t}=\log N_{0}-\frac{t}{D}$$
where *N_t_* is the bacterial population at time *t* (CFU/g), *N*_0_ is the initial bacterial population (CFU/g), and *D* is the decimal reduction time (s) at a specific treatment temperature.

OriginPro 8.1 software (OriginLab, Northampton, MA, USA) was used for model fitting. Fitting goodness of the model was characterized by the correlation coefficient (pseudo-*R*^2^) and the root mean square error (RMSE). The error criterion chosen for this study was the standard error of prediction (SEP) expressed as a percentage and has the advantage of being dimensionless. SEP was calculated by the following expression:(4)$$SEP=\frac{100}{\bar{y}}\sqrt{\frac{{\sum_{i=1}^{N}\left(y^{\prime }-y\right)}^{2}}{N}}$$
where *y’* is the predicted value, *y* is the observed value, 
$\bar{y}$ is a mean of the observed value and *N* is the number of trials. The value of SEP ranging from 0–35% is within the typical plate count error [26].

### 2.7. Statistical Analysis

The results of bacterial populations on samples were analyzed by calculating the means and standard deviations with Excel 2010 (Microsoft, Redmond, WA, USA). ANOVA was performed using least squares techniques with IBM SPSS Statistics 20 software (SPSS, Chicago, IL, USA). A significant difference was established at *p* < 0.05.

### 2.8. Color Measurements

Color changes of chicken wingettes that occurred during scalding were determined using a Chroma Meter CR 400 instrument (Minolta, Osaka, Japan). Skin injuries would happen during scalding, leading the skinless chicken meat exposure to the hot water. Color changes of skinless chicken breast were also determined during scalding. Samples were measured every 10 s within a 100 s treatment. Values of *L*, *a* and *b* representing the lightness, redness and yellowness, respectively, were recorded. The total color difference (Δ*E*) was calculated according to Equation (5). Δ*E* values of 1 to 2 mean that color change is perceived through close observation, and values of 3 to 10 mean changes is perceived at a glance [27]. All measurements were taken on five sites of each chicken sample.
(5)$$\Delta E=\sqrt{\Delta L^{2}+\Delta a^{2}+\Delta b^{2}}$$
where Δ*L*, Δ*a*, and Δ*b* are the differences of lightness, redness, and yellowness between the treated and untreated samples, respectively.

### 2.9. Transmission Electron Microscopy (TEM)

In order to reconfirm the destruction of the *Salmonella* cells during scalding, ultrastructural changes of bacterial cells were observed with a transmission electron microscope (TEM, Hitachi 7650, Ibaraki, Japan). The treated samples were removed from the water bath and placed into a sterile stomacher bag containing 25 mL BPW and then squeezed by hand to wash off attached bacteria. The wash solution for each treatment was collected into a 50 mL centrifuge tube and centrifuged at 6000 × g for 10 min. The bacterial pellets were transferred into sterile 1.5 mL centrifuge tubes. Samples were fixed with 1 mL of 2.5% (v/v) glutaraldehyde (Sangon Biotech, Shanghai, China) and examined using a TEM as previously described [28].

## 3. Results and Discussion

### 3.1. Bacterial Reduction in Scalding

Bacterial reductions on chicken breasts and wingettes at the end of 100 s washing treatments were 0.24 ± 0.13 and 0.28 ± 0.15 log CFU/g at 25 °C, respectively (Figure 2). As shown in Figure 2A, the reductions of *Salmonella* on chicken breasts at the end of 100 s treatments were 1.12 ± 0.07, 1.38 ± 0.01, and 2.17 ± 0.11 log CFU/g at 50, 60 and 70 °C, respectively. There were no significant differences of bacterial reductions between scalding at 50 and 60 °C, and the treatment time was not a significant variable (*p* > 0.05). At the scalding temperature of 70 °C, the bacterial population declined with the treatment time (*p* < 0.05). For chicken wingettes, 0.87 ± 0.02, 0.99 ± 0.14 and 1.11 ± 0.17 log CFU/g reductions were obtained at 50, 60 and 70 °C, respectively (Figure 2B). *Salmonella* on chicken wingettes was less sensitive than that on chicken breasts, especially at 70 °C. No significant differences of bacterial reductions were found among three temperatures, and the treatment time was not a significant variable as well (*p* > 0.05).

According to the sample weight and size, the density of the chicken breasts and wingettes were 0.83 and 0.84 g/cm^3^, respectively. The chicken breasts almost have the same density and thickness as the chicken wingettes. Besides the sample density, various factors have been determined to be involved with bacterial reductions, including the temperature, physical-chemical properties of the surface material, fat content, and treatment time [19,20,27,29,30,31]. As shown in Figure 3, when scalding with water temperatures of 50 and 60 °C, the surface temperatures of chicken breasts and wingettes were below 46 and 45 °C, respectively. The highest temperatures at which *Salmonella* can grow are 42–50 °C, suggesting that the hot washing treatment at 50 and 60 °C could not cause bacteria death [32]. However, bacterial reductions at 50 and 60 °C were greater than washing at 25 °C (Figure 2). The reason could be that the bacteria on the surface immediately in contact with hot water may be killed. McCormick et al. (2003) [33] found that a reduction of *Salmonella typhimurium* on turkey meat was approximately 1.1 log CFU/g at 60 °C. Yang et al., (2001) [19] reported that when scalding temperature increased from 50 to 60 °C, the bacterial reduction of *Salmonella* on chicken skins was 2 log CFU/cm^2^. In this study, bacterial reductions on chicken breasts and wingettes were less than 2 log CFU/g at 50 and 60 °C, showing the effect of temperature on reduction of *Salmonella* was limited, which was consistent with the previous studies. Considering the low of bacterial reductions at 50 and 60 °C within the 100 s treatments, a longer exposure time (>100 s) or hotter water at 70 °C could be used in poultry scalding.

At the scalding temperature of 70 °C, the reduction of *Salmonella* on chicken breast was more significantly effective than on chicken wingette (*p* < 0.05). One reason could be that skinless chicken breasts had a lower fat content compared to the skin-on chicken wingettes. It has been reported that on high fat foods, the bacterial *D*-values were four to eight times higher than those on low fat foods [33,34]. Juneja et al. (2001) [30] studied the reduction of *Salmonella* at temperatures ranging from 58 to 65 °C in beef (12.5% fat), turkey (9% fat), and chicken (7% fat), and found that higher fat concentrations resulted in higher *D* values for *Salmonella* in these products. Besides, the meat microtopography has a significant effect on the bacterial decontamination. Chicken skin depends to a large extent on surface roughness, and the thermal inactivation efficacy decreases as the roughness increases. The reason behind the observed inactivation trend was thought to be that bacteria attach themselves to complex surface features on rougher surfaces and escape decontamination treatment, thus a lesser microbial inactivation was seen on rougher surfaces [35].

### 3.2. Statistical Distributions for Bacterial Reduction at 50, 60 and 70 °C

No significant differences of the bacterial reductions with scalding times were found on chicken breasts at 50 and 60 °C and on chicken wingettes 50, 60 and 70 °C (*p* > 0.05). Besides, there were no significant differences in bacterial reductions at 50 and 60 °C on chicken breasts, as well as at 50–70 °C on chicken wingettes. A logistic (−1.12, 0.06) distribution was chosen based on the result of the Kolmogorov–Smirnov test to describe the bacterial reductions on chicken breasts at 50–60 °C within 100 s treatment (Figure 4A). For chicken wingettes, a logistic (−0.95, 0.07) distribution could describe the bacterial reductions at 50–70 °C (Figure 4B). Figure 4 provided a distribution of all potential results (e.g. different levels of bacteria can be obtained with associated probabilities of occurrence). Although inactivation kinetic parameters are often assumed to follow a normal distribution, normal-like distributions are rarely observed and thus care must be taken when choosing a statistical distribution [36,37]. Alternative and commonly used distributions such as beta, logistic, and exponential distributions can describe better the variability and uncertainty of the data [36,37,38].

### 3.3. Model Evaluation, Comparison and its Application in Quantitative Microbial Risk Assessment (QMRA)

Significant differences in the bacterial reductions and scalding times were found on chicken breasts at 70 °C (*p* < 0.05). Parameters and goodness-of-fit indices (pseudo-*R*^2^, RMSE, SEP) for the Weibull, exponential and log-linear models are shown in Table 1, and the fitting lines are plotted in Figure 5. Compared to the exponential and log-linear models, the Weibull model showed the best fit as indicated by the smaller pseudo-*R*^2^, RMSE and SEP values. The value of the shape parameter *α* at 70 °C was <1, agreeing with the concave upward reduction curve and suggesting that the remaining cells have the ability to adapt to the applied stress. The Weibull model assumes that the heat resistance of bacterial cells is different and the reduction curves exhibit a cumulative form of the distribution of lethal events. This is more accurate to describe the shoulder effect of concave downward or upward curves [23,24]. A paradigmatic study showed that the Weibull model has been selected to describe the bacterial inactivation data at 60, 70, 80 and 90 °C [24]. Similarly, Yang et al. (2002) [20] chose the Weibull distribution model for predicting the survival of *S.typhimurium* and *C. jejuni* during poultry scalding as the primary model. Ground beef inoculated with a 4-strain cocktail of *L. monocytogenes* was subjected to heating at 57, 60, 63, and 66 °C to develop isothermal kinetic models. Experimental data showed that the Weibull-type model was more accurate to describe the survival curves compare to the modified Gompertz and linear models [39].

The models developed in lab-scale test should be validated in the pilot scale test before use in the QMRA model. It is worth noting that although predictive models developed for the description of growth/death of *Salmonella* spp. in a laboratory have been used in *Salmonella* exposure assessment [40,41], the potential limitation of the current model is its reliability. For applications in QMRA, the developed model needs to be evaluated for its ability to extrapolate to other strains, other initial contamination level of the pathogen and other poultry products (e.g., chicken carcasses). Therefore, the predictive models embedded in a risk model should be further validated in the pilot scale test, in addition to the ability to fit the microbiological dataset.

### 3.4. Color Changes during Scalding

The total color differences (Δ*E*) of treated chicken breasts and wingettes were calculated (data not shown). There were no obvious color changes of chicken breasts and wingettes after scalding at 50 and 60 °C (Δ*E* < 10). At 70 °C, the *L* values of treated chicken breasts and chicken wingettes increased slightly, showing a brighter appearance. The lightness (*L*) values are used as an indicator of poultry breast meat quality and for evaluating the incidence of pale, soft, and exudative (PSE)-like conditions [42]. In general, the color changes of chicken breasts and chicken wingettes were acceptable based on the visual inspection.

### 3.5. Morphological Changes Revealed by TEM

We closely examined bacterial cell destruction using TEM. *Salmonella* cells in the absence of scalding treatment retained their round shapes and cell wall morphology. The nuclei and intracellular components were evenly distributed within the cytoplasm as expected from healthy cells (Figure 6A,C). However, after scalding the cell walls were rough with expanded membranes and decomposing inner structures. Some cells were completely disrupted and leaked cytoplasmic material to the extracellular medium. Therefore, scalding treatments resulted in high levels of damage to *Salmonella* cells at high water temperature (Figure 6B,D). Heat treatment at 56 °C caused significant membrane damage to *Salmonella entertitidis* [43]. The cell wall and the outer membrane could change their functionality in response to the change in growth temperature. The degree of saturation of membrane fatty acids increased with increasing growth temperature as a means of maintaining a constant degree of fluidity within cell membranes, and the high membrane fluidity also could increase bacterial reduction in response to the increased growth temperature [29,44,45].

## 4. Conclusions

The parameters of scalding treatments in lab-scale tests, including the material ratio, water temperature, and scalding time, were investigated according to the practical conditions in poultry processing plants in China. Greater bacterial reductions of inoculated *Salmonella* were observed on chicken breasts than that on chicken wingettes during scalding at the temperatures of 50–70 °C within 100 s. A logistic (−1.12, 0.06) distribution and the developed Weibull model could describe the bacterial reductions on chicken breasts at 50–60 °C and 70 °C, respectively. For chicken wingettes, a logistic (−0.95, 0.07) distribution could be used to describe the bacterial reductions at 50–70 °C. For preventing the overestimation of bacterial reduction, the fitted logistic distribution associated with bacterial reduction on chicken wingettes was more suitable for use as an input parameter in the QMRA model of *Salmonella* in the poultry supply chain in China.

## Figures and Tables

**Figure 1 microorganisms-07-00165-f001:**
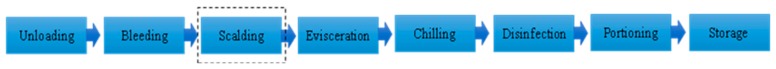
Flow chart of the poultry slaughtering chain. The dashed box is the step of scalding in the poultry slaughtering chain.

**Figure 2 microorganisms-07-00165-f002:**
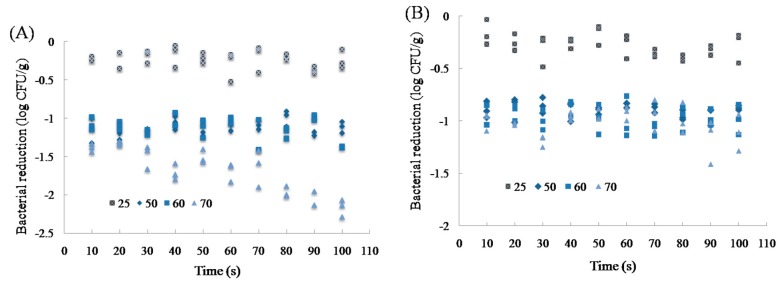
Bacterial reduction of *Salmonella* on chicken breast (**A**) and chicken wingette (**B**) at 25, 50, 60 and 70 °C.

**Figure 3 microorganisms-07-00165-f003:**
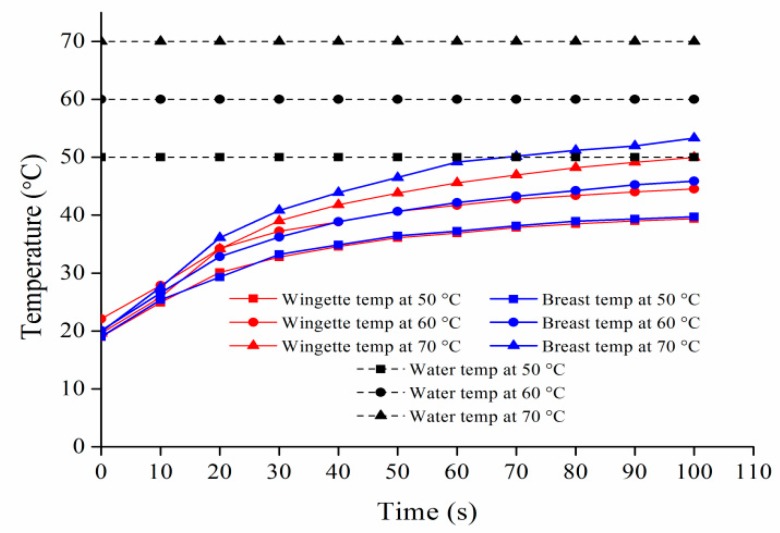
Temperature profiles of chicken breast and chicken wingette immersed in hot water for 100 s at 50, 60 and 70 °C.

**Figure 4 microorganisms-07-00165-f004:**
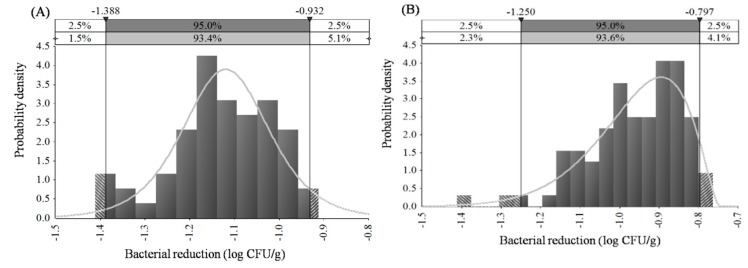
Probability distribution of the reduction of *Salmonella* on chicken breast (**A**) at 50–60 °C and chicken wingette (**B**) at 50–70 °C.

**Figure 5 microorganisms-07-00165-f005:**
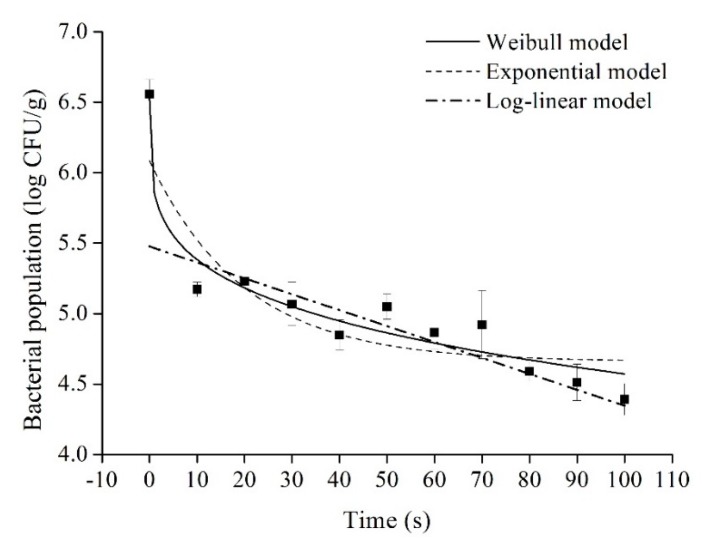
The fitted curves of Weibull, exponential and log-linear models for *Salmonella* reduction on chicken breast at 70 °C. Data points represent the observed bacterial reduction.

**Figure 6 microorganisms-07-00165-f006:**
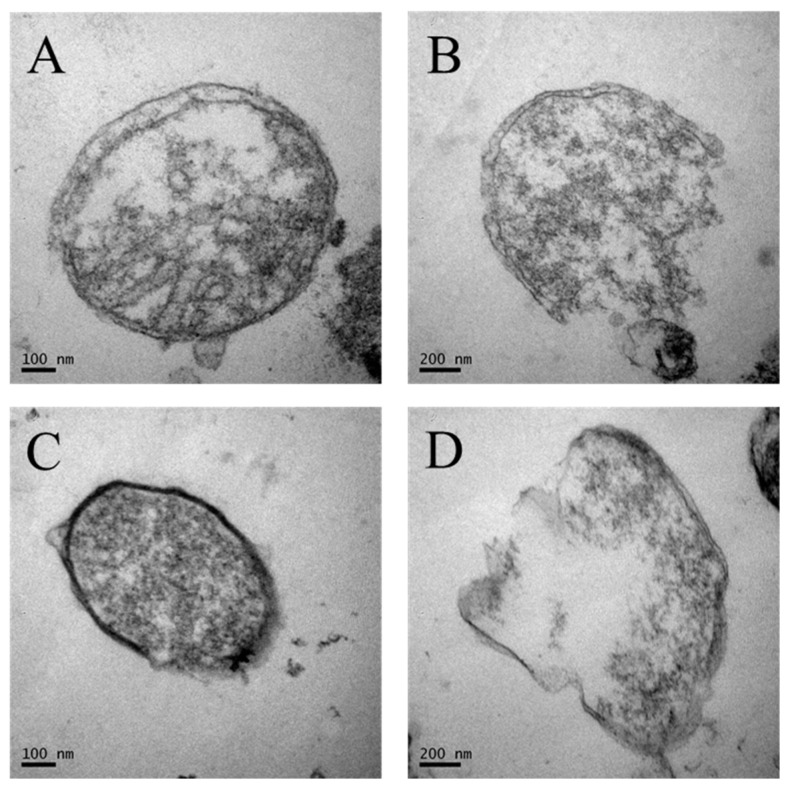
Transmission electron micrographs of *Salmonella* exposed to water at 25 °C for 80 s (**A**); 60 °C for 80 s (**B**); 25 °C for 40 s (**C**); 70 °C for 40 s (**D**).

**Table 1 microorganisms-07-00165-t001:** Parameter estimates and statistical analysis of the Weibull, exponential and log-linear models fitted to *Salmonella* reduction curves on chicken breast at 70 °C.

Weibull Model						Exponential Model						Log-Linear Model				
*N* _0_	*β*	*α*	pseudo-*R*^2^	RMSE	SEP%	*N* _0_	*b*	*c*	pseudo-*R*^2^	RMSE	SEP%	*N* _0_	*D*	pseudo-*R*^2^	RMSE	SEP%
6.5	5.52	0.23	0.90	2.0	2.81	4.7	1.4	20	0.66	3.4	4.7	5.5	88	0.58	3.8	6.9

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
