# Peer review of "Modeling the Reduction of Salmonella spp. on Chicken Breasts and Wingettes during Scalding for QMRA of the Poultry Supply Chain in China"

_microorganisms, 2019, doi:10.3390/microorganisms7060165_

Round 1

Reviewer 1 Report

General comments:

This article is very interesting and well-written describing many aspects of a scalding study, including risk assessment models. However, some clarifications are requested.

In the Introduction, I suggest explaining in short, why these two poultry products were chosen for a scalding experiment (for example add in line 68-75). In line 48 it is expressed that scalding is used for removal of feathers and in line 322 it is expressed that the results will contribute in poultry supply chain in China. Clarify why skinless breasts are used in a scalding experiment. (Experiments with skinless breasts are more related to time/temperature decontamination studies, and then longer exposure time (> 100 s) and hotter water should have been included.) Skin injuries are also related to scalding studies, but this topic is not added in the text.

More precise description: Inoculation of Salmonella spp. should be expressed clearly in the start (line 25-26), since the reduction of inoculated high starting levels of bacteria usually are larger than reductions of naturally contaminated carcasses/products.

Also, add in the abstract or Introduction that this is a laboratory model study (not testing scalding in commercial production)

Details:

Line 26. scalding process: clarify that this is a laboratory experiment using inoculation of Salmonella. (unlike a study with naturally contaminated carcasses in a commercial abattoir).

Line 56. Explain a bit more detailed about In China, due to the difference of facility scale and age of broilers…”. Are the broilers smaller? How small? Cf. the light weight described in line 93, 25 g breasts: are they whole breasts or bits of breasts?

Line 57. Delete are (..temperatures vary from..)

Line 68. Rephrase the line and delete which. (ie. while QMRA of Salmonella were not available in China.)

Line 68. Delete the or replace by previously (In previously published laboratory scale tests..).

Line 72. Delete whether and could be.

Line 97. N=? how many were tested?

Line 108: How was the temperature of chicken meat measured? Was it on the surface or inside the products?

Line 113. Describe the samples used in the analyses, x ml? number of samples?

Line 122: Delete the in preliminary test, since it has not been described in the manuscript before, replace by a. Also, add why you tested on TSA (something about resuscitation step for stressed, sub-lethal bacteria).

N=? how many samples were tested with TSA? (very good that you tested with TSA to check whether there were heat injured bacteria that will grow after adjusting.)

Line 132. Suggest Presently,…” in stead of In nowadays,…”

Line 178-197. Suggest adding a comment about skin injuries of the wingettes?

Line 222 or later. Discuss/add comment about the low bacteria reduction for commonly applied scalding in China, using 50-60 C and up to 100 s. Do you recommend longer scalding time for the slaughterhouses? Higher temperatures? Problems concerning possible skin injuries?

Line 315: Replace 4. Discussion with 4. Conclusion. Suggest also adding inoculated Salmonella and laboratory model study to be more precise in the conclusion.

Author Response

Reviewer 1:

General comments:

This article is very interesting and well-written describing many aspects of a scalding study, including risk assessment models. However, some clarifications are requested.

In the Introduction, I suggest explaining in short, why these two poultry products were chosen for a scalding experiment (for example add in line 68-75).

In line 48 it is expressed that scalding is used for removal of feathers and in line 322 it is expressed that the results will contribute in poultry supply chain in China. Clarify why skinless breasts are used in a scalding experiment. (Experiments with skinless breasts are more related to time/temperature decontamination studies, and then longer exposure time (> 100 s) and hotter water should have been included.) Skin injuries are also related to scalding studies, but this topic is not added in the text.

Reply: “Experiments with skinless chicken breasts are more related to time or temperature decontamination studies. While skin injuries will happen during scalding, leading the skinless chicken meat exposure to the hot water”, the sentence has been added in lines 76-78 to clarify why skinless breasts are used in a scalding experiment.

More precise description: “Inoculation” of Salmonella spp. should be expressed clearly in the start (line 25-26), since the reduction of inoculated high starting levels of bacteria usually are larger than reductions of naturally contaminated carcasses/products.

Also, add in the abstract or Introduction that this is a laboratory model study (not testing scalding in commercial production)

Reply: As suggested, “inoculated” has been added before “Salmonella” in line 22 and “in lab-scale test” has been added in the Introduction (line 83).

Details:

Line 26. “scalding process”: clarify that this is a laboratory experiment using inoculation of Salmonella. (unlike a study with naturally contaminated carcasses in a commercial abattoir).

Reply: The sentence has been revised to clarify this (lines 22, 83).

Line 56. Explain a bit more detailed about “In China, due to the difference of facility scale and age of broilers…”. Are the broilers smaller? How small? Cf. the light weight described in line 93, 25 g breasts: are they whole breasts or bits of breasts?

Reply: Yellow-feathered broiler and white-feathered broiler are two of the most popular chicken species in China. Compare to white-feathered broiler, yellow-feathered broiler with a yellow skin is easily affected by temperature during scalding. Generally, the scalding water temperatures of yellow-feathered broiler vary from 50 to 60 °C and the scalding water temperatures white-feathered broiler vary from 60 to 70 °C according to the onsite investigation. The sentence has been added in lines 54-58. The 25 g breasts are bits of breasts.

Line 57. Delete “are” (..temperatures vary from..)

Reply: “are” has been deleted in line 53.

Line 68. Rephrase the line and delete “which”. (ie. …while QMRA of Salmonella were not available in China.)

Reply: It has been rephrased and the sentence was changed to “…while these data are not available for the QMRA of Salmonella in China” in line 71.

Line 68. Delete “the” or replace by “previously” (In previously published laboratory scale tests..).

Reply: “the” has been replaced by “previously” (line 71).

Line 72. Delete “whether” and “could be”.

Reply: “whether” and “could be” have been deleted (line 75)

Line 97. N=? how many were tested?

Reply: Three chicken breasts and three wingettes were tested in control experiments. The sentence has been revised as “Control experiments showed that no Salmonella was initially present on the three chicken breasts and three wingettes, respectively” in line 102.

Line 108: How was the temperature of chicken meat measured? Was it on the surface or inside the products?

Reply: The thermocouples (34970A, Agilent, Santa Clara, CA) were inserted on the top surface of samples to measure the surface temperature (lines 124-125).

Line 113. Describe the samples used in the analyses, x ml? number of samples?

Reply: The samples were individually added to sterile stomacher bags containing 25 mL buffered peptone water and homogenized for 1 min in a Model 400 food stomacher (lines 114-116). The homogenates were serially 10-fold diluted in BPW and a 50 µL portion of appropriate dilutions was plated in duplicate onto the XLT4 agar using a spiral plater (lines 133-135). Thirty inoculated chicken breasts were submerged into the water bath and three samples were removed for processing every 10 s within a 100 s treatment period. Each treatment was repeated three times (lines 113-114, 121).

Line 122: Delete “the” in “preliminary test”, since it has not been described in the manuscript before, replace by “a”. Also, add why you tested on TSA (something about resuscitation step for stressed, sub-lethal bacteria).

N=? how many samples were tested with TSA? (very good that you tested with TSA to check whether there were heat injured bacteria that will grow after adjusting.)

Reply: “In the preliminary test” has been revised to “In a preliminary test” in line 128. To clarify why tested on TSA agar, the sentence has been added in lines 128-131.

Line 132. Suggest “Presently,…” in stead of “In nowadays,…”

Reply: “In nowadays” has been revised to “Presently” in line 140.

Line 178-197. Suggest adding a comment about skin injuries of the wingettes?

Reply: As suggested, “Color changes of chicken wingettes that occurred during scalding were determined using a Chroma Meter CR 400 instrument (Minolta, Osaka, Japan). Skin injuries would happen during scalding, leading the skinless chicken meat exposure to the hot water. Color changes of skinless chicken breast were also determined during scalding” has been added in lines 184-187.

Line 222 or later. Discuss/add comment about the low bacteria reduction for commonly applied scalding in China, using 50-60 C and up to 100 s. Do you recommend longer scalding time for the slaughterhouses? Higher temperatures? Problems concerning possible skin injuries?

Reply: Considering the low bacterial reductions at 50 and 60 °C within the 100 s treatments, a longer exposure time (> 100 s) or hotter water at 70 °C could be used in poultry scalding (lines 233-234).

Line 315: Replace “4. Discussion” with “4. Conclusion”. Suggest also adding “inoculated” Salmonella and “laboratory model study” to be more precise in the conclusion.

Reply: “Discussion” has been revised to “Conclusion” in line 321 and the “Conclusion” has been revised to make it more precise (lines 322-323). 

Reviewer 2 Report

Referee for Microorganisms, paper ID 516297

The reviewed paper is a research paper aiming at modelling the reduction of Salmonella contamination on chicken breasts and wingettes during scalding. Different time/temperatures couples usually used in China have been tested.

This paper is interesting as it deals with a control measure of human Salmonella infections originating from poultry, through experiments conducted in conditions mimicking the scalding environment. The paper is well written, with an interesting approach and nice results.

Globally from my point of view the paper should be considered for publication after answers to my questions and some minor revisions, detailed hereafter.

My major problem resides in the fact that there is no description of the slaughtering chain with all the steps: therefore, it is difficult to appreciate if the presented model is relevant. This chain emphasising the scalding step could be presented as a supplementary material. I particularly wonder if scalding is applied in China to breasts or wingettes after cutting of the carcass, of if it is applied to the whole carcass. This could lead to significantly different results. Moreover, is the scalding step mandatory in all Chinese abattoirs or are there different processes?

What were the criteria to choose the 5 Salmonella strains? Heat resistance features? Were they isolated from scalding baths?

Even if I worked on QRA, I do not feel competent on the choice of the mathematical approaches. I feel however they sound good for me.

Line 286: I was a bit confused by the fact that the authors evoke robustness as a potential limitation. Is this OK? can this be explained?

Throughout the whole text, titles have capital letters for all words: is this the journal’s policy? Please verify this point.

Line 25: “The” should not be bold.

Lines 28-29: it could be stressed that the same time (100s) was applied for the experiment on wingettes.

Line 40: Salmonella is pathogenic also for many animal species

Line 57: “are vary” should read “vary”

Line 67: “had” should read “have”

Line 68: I suggest replacing “which were” with “these date are”

Line 83: I suggest replacing “Five strains of Salmonella” with “Five Salmonella strains belonging to different serotypes”

Lines 88, 111, 126, 130, 193, 194, 198 and 196: “ml” and µl” should read “mL” and “µL”, respectively.

Lines 100-101: I understand these results come from the determination presented lines 113-114. This should be underlined, and I would find more relevant to present the results after lines 113-114.

Line 138: I suggest replacing “that” with “for which”

Line 143: please add “a” between “when” and “significant”

Lines 220-222: I suggest replacing “Due to the highest temperatures at which Salmonella can grow were 42-50 °C” with “The highest temperatures at which Salmonella can grow are 42-50 °C”

Line 223: please remove “that” before “washing”

Lines 225, 275 and 307: names of serotypes should be in roman with a capital letter (see lines 83-84!), i.e. Typhimurium and Entertitidis.

Line 250: please add “on” after “based”

Lines 286-288: I suggest replacing “For applications in QMRA, the developed model needed to evaluate the ability of the developed model to extrapolate to other strains” with “For applications in QMRA, the developed model needs to be evaluated for its ability to extrapolate to other strains”

Line 299: “Reveled” should read “Revealed”

Line 315 “Discussion” should read “Conclusion”

Author Response

Reviewer 2:

Referee for Microorganisms, paper ID 516297

The reviewed paper is a research paper aiming at modelling the reduction of Salmonella contamination on chicken breasts and wingettes during scalding. Different time/temperatures couples usually used in China have been tested.

This paper is interesting as it deals with a control measure of human Salmonella infections originating from poultry, through experiments conducted in conditions mimicking the scalding environment. The paper is well written, with an interesting approach and nice results.

Globally from my point of view the paper should be considered for publication after answers to my questions and some minor revisions, detailed hereafter.

My major problem resides in the fact that there is no description of the slaughtering chain with all the steps: therefore, it is difficult to appreciate if the presented model is relevant. This chain emphasising the scalding step could be presented as a supplementary material. I particularly wonder if scalding is applied in China to breasts or wingettes after cutting of the carcass, of if it is applied to the whole carcass. This could lead to significantly different results. Moreover, is the scalding step mandatory in all Chinese abattoirs or are there different processes?

Reply: According to the operate procedure of chicken slaughtering (GB/T 19478-2004), flow chart of the poultry slaughtering chain has been added in the Introduction (lines 44-45). Scalding is applied to the whole carcass after bleeding. While skin injuries would happen during scalding, leading the skinless chicken meat exposure to the hot water. Therefore, skinless chicken breast and shin-on wingette were considered in this study. Scalding step is mandatory in poultry slaughtering in China according to the operate procedure of chicken slaughtering.

What were the criteria to choose the 5 Salmonella strains? Heat resistance features? Were they isolated from scalding baths?

Reply: The five Salmonella serotypes were dominant strains isolated from chicken carcasses. For clarify this, the sentence has been revised in lines 88-91.

Even if I worked on QRA, I do not feel competent on the choice of the mathematical approaches. I feel however they sound good for me.

Line 286: I was a bit confused by the fact that the authors evoke robustness as a potential limitation. Is this OK? can this be explained?

Reply: “robustness” has been revised to “reliability” (line 293). Reliability of predictions should be evaluated for its ability to extrapolate to other strains, other initial contamination level of the pathogen and other poultry products (e.g., chicken carcasses)

Perez-Rodriguez, F.; Valero, A. Predictive microbiology in foods. In Predictive microbiology in foods 2013, (pp. 1-10). Springer, New York, NY.

Throughout the whole text, titles have capital letters for all words: is this the journal’s policy? Please verify this point.

Reply: The titles in the whole text are capital letters according to the journal’s policy.

Line 25: “The” should not be bold.

Reply: It has been revised in line 22.

Lines 28-29: it could be stressed that the same time (100s) was applied for the experiment on wingettes.

Reply: As suggested, the sentence has been revised as “For chicken wingettes, 0.87 ± 0.02, 0.99 ± 0.14 and 1.11 ± 0.17 log CFU/g reductions were obtained at 50, 60 and 70 °C after the 100 s treatments, respectively” in lines 25-26.

Line 40: Salmonella is pathogenic also for many animal species

Reply: The sentence has been revised as “Salmonella spp. are Gram-negative facultative intracellular zoonotic pathogens” in line 37.

Line 57: “are vary” should read “vary”           

Reply:  “are” has been deleted (line 53).

Line 67: “had” should read “have”

Reply:  “had” has been revised to “have” in line 70.

Line 68: I suggest replacing “which were” with “these date are”

Reply: “which were” has been revised to “these date are” in line 71.

Line 83: I suggest replacing “Five strains of Salmonella” with “Five Salmonella strains belonging to different serotypes”

Reply: As suggested, “Five strains of Salmonella” has been revised as “Five Salmonella strains belonging to different serotypes” (line 88).

Lines 88, 111, 126, 130, 193, 194, 198 and 196: “ml” and µl” should read “mL” and “µL”, respectively.

Reply: “ml” and µl” have been revised in lines 93,115,134,137,198-201 and also throughout the revised manuscript.

Lines 100-101: I understand these results come from the determination presented lines 113-114. This should be underlined, and I would find more relevant to present the results after lines 113-114.

Reply: The sentence in lines 105-106 has been moved to lines 117-118.

Line 138: I suggest replacing “that” with “for which”

Reply: As suggested, “that” has been revised to “for which” in line 145.

Line 143: please add “a” between “when” and “significant”

Reply: “a” has been added before “significant” in line 150.

Lines 220-222: I suggest replacing “Due to the highest temperatures at which Salmonella can grow were 42-50 °C” with “The highest temperatures at which Salmonella can grow are 42-50 °C”

Reply: The sentence has been revised as suggested in lines 223-224.

Line 223: please remove “that” before “washing”

Reply: “that” has been deleted in line 226.

Lines 225, 275 and 307: names of serotypes should be in roman with a capital letter (see lines 83-84!), i.e. Typhimurium and Entertitidis.

Reply: The names of serotypes have been revised in lines 228, 313, 352. And also in lines 358, 389,393, 418, 431, 443-444, 460.

Line 250: please add “on” after “based”

Reply: “on” has been inserted after “based” in line 254.

Lines 286-288: I suggest replacing “For applications in QMRA, the developed model needed to evaluate the ability of the developed model to extrapolate to other strains” with “For applications in QMRA, the developed model needs to be evaluated for its ability to extrapolate to other strains”

Reply: The sentence has been revised as suggested in lines 293-294.

Line 299: “Reveled” should read “Revealed”

Reply: Reveled” has been revised to “Revealed” in line 306.

Line 315 “Discussion” should read “Conclusion”

Reply: Discussion” has been revised to “Conclusion” in line 321
